# Reinforcement Analysis of an Old Multi-Beam Box Girder Based on a New Embedded Steel Plate (ESP) Strengthening Method

**DOI:** 10.3390/ma15124353

**Published:** 2022-06-20

**Authors:** Yuliang He, Kai Wang, Zongyong Cao, Peijuan Zheng, Yiqiang Xiang

**Affiliations:** 1College of Civil Engineering, Shaoxing University, 508 West Huangcheng Road, Shaoxing 312000, China; hyliang88888@163.com (Y.H.); wangkai1999622@163.com (K.W.); 2Hua Hui Group, Shaoxing University, 508 West Huangcheng Road, Shaoxing 312000, China; cao-zy@cnhh.com; 3College of Civil Engineering and Architecture, Zijingang Campus, Zhejiang University, Anzhong Building B718, Yuhangtang Road 866, Hangzhou 310058, China; xiangyiq@126.com

**Keywords:** multi-beam box girder bridge, ESP strengthening, field test, load lateral transferring performance, structural stiffness

## Abstract

Multi-beam box girder bridges have been applied widely throughout the world for many years. However, the cracking of longitudinal joints between the box girders always leads to reflective cracking of the bridge decks during the service period and thus finally affects the safety and durability of the actual bridges. An embedded steel plate (ESP) strengthening method was presented by introducing carbon-A/-B glue to reinforce the longitudinal joints of old multi-beam box girder bridges for this problem. In order to evaluate the feasibility of the proposed method for actual bridges, an old multi-beam box girder bridge was reinforced, and structural parameters including strain, frequency, and deflection were obtained by adopting field tests before and after strengthening. In addition, the corresponding finite element (FE) model of the background bridge was also set up using ANASYS 18.0 to analyze the strengthening process. Analysis results of the actual bridge and FE model indicate that structural stiffness and load lateral transferring performance between the box girders were enhanced after ESP strengthening. Therefore, this proposed strengthening method can be used to improve the mechanical performance of multi-beam box girder bridges and provide reference for such bridge reinforcement.

## 1. Introduction

Multi-beam box girder bridges have always been applied in short- and medium-span bridges. Those bridges were longitudinally connected together through full-depth and partial-depth shear keys between adjacent girders to enhance the bridge’s integrity [1]. Due to accelerated bridge construction, high torsional stiffness, favorable depth-to-span ratios, cost-effectiveness, and aesthetic appeal [2,3], multi-beam box girder bridges are one of the most popular bridges and have been applied widely for many years in the world. However, this type of bridge has lost attractiveness little by little because of the longitudinal cracks at the girder-key interface or in the shear key that are created under a combination of live loads and temperature [4]. These cracks can spread to the bridge slab surface and lead to salt and water leakage through the shear key, and they can accelerate the corrosion of the joint rebar. Lateral load transferring among the box girders can be lost, which leads to safety concerns [5]. Attanayake and Aktan [2] tried to apply a cast-in-place concrete slab, transverse post-tensioning, full-depth shear keys, and seven-day moist curing of the bridge slab to mitigate reflective cracks. The results showed that these methods failed to solve the longitudinal joint cracking of multi-beam box girder bridges.

In America, four types of keyway details (three generic partial-depth shear keys and a generic narrow full-depth shear key) are used. Conversely, the wide full-depth shear key is applied in South Korea, Japan, and England [6,7,8]. El-Remaily et al. [9] indicated that longitudinal cracks were seldom found in multi-beam box girder bridges with wide full-depth shear keys. At present, four grouting materials (shrinkage-compensated concrete, fiber-reinforced concrete, cement-based construction grout, and epoxy grout) were applied in the shear key. It is easy to de-bond at the interface of the shear key, which is filled with cement-based construction grout, during the early-age period when the shear key is subjected to the heat of hydration and joint material expansion. Liu and Phares [10] showed that the wide full-depth joint filled with shrinkage-compensated concrete performed better than the narrow full-depth joint filled with epoxy grout. It was also found that the wide full-depth joint filled with shrinkage-compensated concrete would have cracks during the early-age period. However, the epoxy has a high thermal expansion relative to that of the concrete, and this may lead to cracking of the epoxy grout. Sang [11] found that longitudinal cracks have not occurred in the full-depth shear key but did occur in the partial-depth keyways using fiber-reinforced cementitious grout. This showed that it was very important to choose the grouting material and joint details to mitigate longitudinal cracking of the bridge slab. Yuan and Graybeal [12] reported a new shear key design for multi-beam box girder bridges using ultra-high performance concrete (UHPC). Hussein and Steinberg [13] found that UHPC might enhance the capacity of the shear key and mitigate reflective cracking of the bridge deck due to high adhesion and higher strength of the UHPC material, and the adhesive strength increased with surface roughness (adhesion/cohesion values of 3.01, 5.01, and 5.63 MPa for smooth, mid-rough, and rough surfaces). Shad et al. [14] and Ali et al. [15] showed that a UHPC joint with a smooth surface and minimum transverse rebar was sufficient for load transfer between the beams for the load level used in the Federal Highway Administration testing, and these research studies indicated that a UHPC joint with a mid-rough surface and minimum transverse rebar was capable of transferring the load up to failure of the girder system.

Multi-beam box girder bridges constitute one-sixth of all the bridges in America [5], and they account for the largest percentage (about 85%) of bridges built during the early years in China. About 73% of these bridges have partial-depth shear keys, about 80% have spans ranging between 12 m and 18 m, and more than 15% are replaced or built each year [16]. The cause for the replacement is that the shear key is ruptured, which leads to the loss of lateral load transfer between the box girders, and the capacity of the single box girder is further reduced under the action of overloaded vehicles and environmental variations, which creates safety concerns. If the shear key is reinforced to enhance the overall performance of box girder bridges, there is no need to rebuild or replace most of the multi-beam box girder bridges with the distress, which can avoid inconvenient vehicle traffic and economic loss. To the authors’ knowledge, few studies were conducted to strengthen the longitudinal joints of multi-beam box girder bridges. Attanayake and Aktan [2] and Shi [17] tried to apply transverse post-tensioning and large diameter screws to reinforce the shear key of box girder bridges, and this required that the stress must be constant per unit length in the direction of the bridge span. In fact, the compressive stress is only limited to a slab width of 1.35 times in the direction of post-tensioning, and it varied with the bridge width, which made the reinforced effect of transverse post-tensioning ineffective. Fu et al. [18] also reported that the transverse post-tensioning had no effect on the load transfer between the girders before the cracking of the shear key.

Since the full-depth shear key with high adhesive strength of the grouting material might mitigate reflective cracking [11,12], the authors tried to propose a new method by introducing a new grouting material (carbon-A/-B glue) to strengthen the partial-depth keyways. Carbon-A/-B glue has outstanding advantages in adhesion/cohesion strength and shear strength, and it especially gives good mobility to grout, conveniently, in the narrow gaps between the box girders. To reduce the amount of carbon-A/-B glue, the steel plates are embedded in the shear key and the gaps between the girders as the skeleton. The box girders and steel plates are pasted by the carbon-A/-B glue in the full depth of the longitudinal joints, as the full-depth shear key, to make the box girder bridge behave monolithically. In this study, an old multi-beam box girder bridge was strengthened using the ESP method. Then its corresponding structural performance before and after ESP strengthening were obtained and analyzed by adopting field tests. Finally, a three-dimensional finite element (FE) model was developed to further investigate the strengthening effect.

## 2. Experimental Program

### 2.1. Bridge Description

The background bridge, located in Shaoxing, China, was constructed in 1994 and was scheduled to be dismantled in 2021. Its three box girders of 1# span were selected as the experiment subject to evaluate the feasibility of the proposed ESP method. Figure 1 is composed of three figures, and these figures show the longitudinal section, girder section, and transverse section, respectively. It can be seen that the bridge consists of five spans and the total length is 86 m. The length of the 1# and 3# span is 13 m and other spans are 20 m. The width of the bridge slab is 23.3 m, and each girder of 1# span is 990 mm and 550 mm in width and depth, respectively, as shown in Figure 1b. Each span of the bridge is made up of twenty-one box girders. Each girder of 1# span is prestressed using nine 9.5-mm-diameter pre-tensioning strands and all these strands are stressed to 195 kN. The box girders are connected together by adopting partial-depth shear keys, which incorporate an on-site concrete slab (100 mm deep) with monolayer rebar mesh. In 2002, the reflective cracking of the bridge slab was observed in the longitudinal joints, and a few cracks were also found in the bottom surface of the box girders. In the following years, the reflective cracking of the bridge slab became worse with the increasing of traffic loads and the overloading of vehicles. In 2006, the partial-depth shear keys and bridge slab were removed, and the longitudinal joints were rebuilt using concrete with a compressive strength of 40 MPa. The bridge deck was replaced by a cast-in-place concrete slab (200 mm deep) with two-layer rebar mesh, and the designed compressive strength of the concrete is also 40 MPa. In 2018, the longitudinal reflective deck crack was observed again, and the damage degree was less than that in 2006. This showed that the increasing of the depth and reinforcement ratio of the bridge slab might relieve the development of reflective cracking. With the service time increasing, the reflective cracking of the bridge slab would become more and more severe, which was consistent with Attanayake and Aktan [2].

### 2.2. Material Properties

It can be seen in the introduction that the full-depth shear key with high adhesive strength of grouting material might mitigate reflective cracking. Since the carbon-A/-B glue, a kind of composite material, has high tensile and shear strength, high adhesion/cohesion strength, and good mobility, it is chosen in our work. The carbon-A/-B glue is shown in Figure 2. The carbon-A/-B glue is divided into an A-type and B-type in storage. When the two kinds of glues are mixed together, the solidification takes place after about six hours. The mixture ratio is A:B = 1:2. According to the product instructions, the adhesion strength of the concrete-concrete interface with a smooth surface is 4.2 MPa; this is larger than that of the HSC-UHPC interfaces with a smooth surface (3.5 MPa) [11], whereas the adhesion strength of the steel-steel interface is 35 MPa. The steel plate was embedded in the shear keys, and the gaps between the box girders might further improve the adhesion strength of the girder-key interface or the girder-girder interface. The elongation of the glue after the solidification is 2.1%, and the shear deformation is less than 0.2 mm. Huckelbridge et al. [15] reported that the shear key is fractured if the relative displacement value is between 0.08 and 0.5 mm, whereas a relative displacement value of 0.025 mm or less between box girders indicates that the shear key remains intact when the grouting material is concrete. This also shows that the carbon-A/-B glue endures a good shear deformation in comparison with concrete. The material properties of the steel plate and carbon-A/-B glue are listed in Table 1.

### 2.3. Schematic Process

Figure 3 shows the details of the steel plate, which is embedded in the shear keys and gaps between the box girders. The steel plate is divided into two parts. One part is welded into the grooved section located in the shear keys, and another part is embedded directly into the gaps between the box girders, as shown in Figure 3a. The PVC pipes are inserted by drilling holes at the bottom of the grooved section to grout the carbon-A/-B glue. The length of the steel plate is more than 100 mm from the bottom surface of the box girder, and some holes were drilled at the exceeding part of steel plate, as shown in Figure 3b. Two angle irons are connected to the steel plates, using the bolts to resist the pressure caused by grouting the carbon-A/-B glue, as shown in Figure 3a. The sealing strips are pasted between the angle irons and the bottom surface of the box girder to prevent leakage of the carbon-A/-B glue. The steel plate is divided into thirteen parts in the longitudinal of the box girder in order to be easily installed, as shown in Figure 3b.

Figure 4 shows the process of the ESP strengthening method. As the initial step, the bridge slab and shear keys of the old bridge are chipped away and then the girder-key interface and girder-girder interface are cleaned up (Figure 4a). The steel plates are embedded in the shear keys and gaps between the box girders, as shown in Figure 3. Two angle irons are placed flat and tight against the underside of the box girders at the gaps between the box bridges and are connected with the steel plate by the bolts, as shown in Figure 3a and Figure 4b. The sealing strips are applied to plug the gaps between the angle irons and the underside of the box girders to prevent leakage of the carbon-A/-B glue (Figure 4b). Some holes need be reserved at the bottom of the grooved steel located in the partial-depth shear keys in order to install the PVC pipes, and the carbon-A/-B glue is injected into the spaces between the steel plate and box girders through the PVC pipes. The upper ends of the spaces between the grooved steel and the box girders are also sealed by the strips to prevent leakage of the concrete when the bridge deck and the shear keys are cast-in-situ (Figure 4c). The carbon-A/-B glue is injected into the spaces between the steel plates and box girders through the PVC pipes until the concrete strength reaches 40 Mpa (Figure 4d). The carbon-A/-B glue was grouted from the mid-span to the edge along the longitudinal joint.

### 2.4. Field Test

A truck was applied to conduct field tests before and after ESP strengthening. Its parameters and configurations are listed in Table 2 and Figure 5, respectively. The tests were proceeded under three cases: the first was continuous loading and the truck was driven along the box bridge, the second was symmetrical loading at the mid-span (Figure 5a), and the third was partial loading at the mid-span (Figure 5b). Figure 6 shows the layout of the strain gauge and LVDTs for the three box girders. Three 1/1000-mm LVDTs were applied to measure the vertical displacements at the mid-span of the box girders (Figure 6), and four 1/1000-mm LVDTs were laid near the gaps to measure the relative displacements between the box girders. Strain gauges were also placed in the underside of the box girders at the mid-span to measure the concrete strain (Figure 6). The data acquisition instrument (DHDAS-3818Y) was used to collect the static experimental data, and all strain gauges and LVDTs were connected to the data acquisition instrument using extension lines (Figure 7). In addition, the dynamic test was also carried out to further verify the reinforcement effect of the ESP method. The frequencies of the box girder bridge were measured under ambient excitation before and after ESP strengthening, and the data logger was used to collect the dynamic test data (Figure 8). Both the dynamic and static loading tests were conducted twice, and the average values were selected as test results.

## 3. Experimental Results

In this study, the ESP method was used to strengthen the longitudinal joint of an old multi-beam box girder bridge. Figure 9 shows the vertical deflections under the symmetrical and partial static loading test before and after ESP strengthening. Under the symmetrical static loading case (Figure 9a), the deflections for the 1#, 2#, and 3# girders were 4.834 mm, 1.085 mm, and 5.018 mm before strengthening, respectively, and 3.766 mm, 3.102 mm, and 3.85 mm after strengthening, respectively. The strengthening effect of the deflections was improved by 28.4%, 185.9%, and 30.3% for the 1#, 2#, and 3# girders, respectively. The relative displacement before strengthening was 0.75 mm between the 1# and 2# girders and 0.93 mm between the 2# and 3# girders. This indicates that the shear keys of the old box girder bridge have been fractured according to Huckelbridge et al. [15]. After strengthening, the relative displacement was 0.02 mm between the 1# and 2# girders and 0.023 mm between 2# and 3# girders. This indicates that the shear keys remain intact. Figure 9b displays the vertical displacements at the mid-span under partial loading before and after strengthening. The displacements were 0.446 mm, 1.553 mm, and 3.437 mm for the 1#, 2#, and 3# girders before strengthening, respectively, and the corresponding displacements were 1.65 mm, 1.692 mm, and 1.965 mm after strengthening, respectively. The displacement of the 3# girder was decreased by 43% and increased by 270% and 9% for the 1# and 2# girders, respectively. The relative displacements before strengthening were 0.51 mm between the 1# and 2# girders and 0.89mm between the 2# and 3# girders. After strengthening, the corresponding relative displacements were 0.012 mm and 0.018 mm. This shows that the whole performance of the box bridge after strengthening was better than that before strengthening.

Figure 10 displays the concrete strain values in the underside of the box girders at the mid-span under the symmetrical and static partial loading tests before and after ESP strengthening. The test results under symmetrical loading presented that the maximum and minimum concrete strain values at the mid-span were 1747 µϵ and 952 µϵ before strengthening, respectively. After strengthening, the corresponding strain values were 1413 µϵ and 1299 µϵ, respectively. Under the case, the maximum concrete strain value of the concrete was decreased by 23.6%, and the minimum concrete strain value was increased by 26.7%. Under the partial loading, the maximum and minimum concrete strain values in the underside of the box girders at the mid-span were 1244 µϵ and 248 µϵ before strengthening, respectively, and 763 µϵ and 637 µϵ after strengthening, respectively. Under the condition, the maximum and minimum concrete strain values were decreased by 63% and increased by 61%, respectively. In one word, the concrete strain values were improved obviously before and after strengthening.

The truck was driven along the box bridge to present the variations in the deflection of the box girder at the mid-span under the symmetrical and partial continuous loading conditions. The deflection curves under the above cases are shown in Figure 11. Before strengthening, the maximum deflection was 5.17 mm under the symmetrical continuous loading case, and the big relative displacement was 0.75 mm. After strengthening, the maximum deflection was reduced to 3.8 mm, and the big relative displacement decreased to 0.021 mm. The maximum deflection was improved by 36%, and the relative displacement met the requirements that Huckelbridge et al. [15] proposed. Under the partial continuous loading case, the maximum deflection before strengthening was 4.44 mm, and the big relative displacement was 0.89 mm. After strengthening, the corresponding values were 2.2 mm and 0.02 mm, respectively. The maximum deflection was improved by 50%, and the relative displacement was less than 0.025 mm. The comparison of the deflections and the relative displacement before and after strengthening indicate that the box girder bridge behaves more monolithically after strengthening.

Figure 12 displays the concrete strain curves in the underside of the box girders at the mid-span under the symmetrical and partial continuous loading tests before and after the ESP strengthening. The test results under the symmetrical continuous loading case presented that the maximum concrete strain value was 1927 µϵ before strengthening, and the corresponding value was 1409 µϵ after strengthening. Under the partial continuous loading case, the maximum concrete strain value was 1542 µϵ before strengthening and 787 µϵ after strengthening. It can be seen that the maximum concrete strain value was decreased by 36.8% under the symmetrical continuous loading case and 49% under continuous partial loading. 

The collection time of the acceleration signal is not less than 30 min in order to get enough data under ambient excitation. Then the signals were used to obtain the Fast Fourier transform (FFT) spectrum of the box bridge (Figure 13). It can be seen that the first three order frequencies before strengthening were 8.875 Hz, 11.274 Hz, and 18.167 Hz, respectively, and the corresponding frequencies were 17.578 Hz, 25.282 Hz, and 30.416 Hz after strengthening, respectively. The fundamental frequency was increased by 46%, and this indicates the overall stiffness of the box girder bridge is improved by adopting the ESP method.

Multi-beam box girder bridges always consist of side-by-side prestressed or precast reinforced concrete box girders, which are then longitudinally connected together using the shear keys. The distribution factor of the vehicle axle load to each box girder is not always equal, as it is commonly characterized by a live load distribution factor. Some researchers have tried to explore the analytical approaches of lateral load distribution by adopting field tests. However, load effects cannot be directly measured in the field, whereas the deflection and strain can be measured. As a result, the distribution factor can alternatively be determined by taking the ratio of the response in a given member to the summation of all primary load-carrying member responses and multiplying the number of trucks applied on the bridge [19], as follows: (1)$$\begin{array}{l} \;g_{i}=\frac{R_{\max,i}}{\sum_{i}^{\mathrm{NO}.girders}R_{\max,i}}\cdot N_{trucks}=\frac{\Delta_{\max,i}}{\sum_{i}^{\mathrm{NO}.girders}\Delta_{\max,i}}\cdot N_{trucks}\\ \begin{array}{cc} & = \end{array}\frac{\varepsilon_{\max,i}}{\sum_{i}^{\mathrm{NO}.girders}\varepsilon_{\max,i}}\cdot N_{trucks} \end{array}$$
where *R*_max_*_,i_* is the maximum reaction force in the *i*th girder; Δ_max_*_,i_* is the maximum deflection in the *i*th girder; *ε*_max,*i*_ is the maximum strain in the *i*th girder; and *N_trucks_* is the number of trucks on the bridge for the given loading.

Figure 14 shows the lateral load distribution factors under the symmetrical and static partial loading cases before and after ESP strengthening. Under the symmetrical static loading case, the deflection distribution factors obtained from Equation (1) before strengthening were 0.44, 0.1, and 0.46 for the 1#, 2#, and 3# girders, respectively, and the coefficient of variation was 0.036. The strain distribution factors were 0.4, 0.22, and 0.38 for the 1#, 2#, and 3# girders, respectively, and the coefficient of variation was 0.039. After strengthening, the corresponding deflection distribution factors were 0.35, 0.29, and 0.36, respectively, and the coefficient of variation was 0.017. The corresponding strain distribution factors were 0.35, 0.32, and 0.33, respectively, and the coefficient of variation was 0.018. Under the case, the coefficient of variation of the deflection and strain distribution factors was improved by 53% and 50% after strengthening, respectively. Under the static partial loading condition, the deflection distribution factors before strengthening were 0.08, 0.29, and 0.63 for the 1#, 2#, and 3# girders, respectively, and the coefficient of variation was 1.167. The strain distribution factors were 0.13, 0.38, and 0.49 for the 1#, 2#, and 3# girders, respectively, and the coefficient of variation was 0.752. After strengthening, the deflection distribution factors were 0.31, 0.32, and 0.37 in order, and the coefficient of variation was 0.126. The corresponding strain distribution factors were 0.31, 0.33, and 0.36, respectively, and the coefficient of variation was 0.12. Under the condition, the coefficient of variation of the deflection distribution factors was improved by 89% and 84% for the strain distribution factor. This indicates that the load lateral transferring performance between the box girders is improved under two cases.

## 4. Finite Element Analysis

To further verify the strengthening effect of the ESP method, ANSYS 18.0 [20] was used to conduct a study based on the background bridge. Figure 15 shows the FE model of the bridge in Section 2.3 and Section 2.4. Concrete box girders, the shear key, and bridge slabs were all simulated as 3-D element SOLID65. Reinforcing bars and pre-tensioning strands were modeled as link elements. Double cells with the irregular figure were converted into the double circular according to the equality of the area and inertia moment. The concrete properties came from Schmidt hammer tests. The elastic modulus of the steel rebar and concrete is 200 GPa and 32.5 Gpa, respectively. Poisson’s ratio of the steel rebar and concrete is 0.3 and 0.18, respectively. The background bridge worked in the elastic behavior under the truck load, which was applied in the field tests, and the non- linearity behavior was not taken into consideration in the FE analysis. Before strengthening, the partial-depth shear key was fractured, and the girder-key interface was thought to be de-bonded in the FE model. After strengthening, because the adhesion strength of carbon-A/-B glue was more than the tensile strength of concrete, the girder-key interface strength of the concrete, the girder-key interface, and the girder-girder interface were thought to be bonded perfectly in the FE model. The loading process was divided into a loading step and a pre-tension step. In the pre-tension step, an initial stress, 195 kN, was applied to all strands. A roller and a hinge were applied as boundary conditions.

In this paper, the Lanczos algorithm was applied for free vibration analysis. The fundamental frequency results are listed in Table 3. The fundamental frequency of the FE analysis was 8.76 Hz and 17.45 Hz before and after ESP strengthening, respectively. The FE analysis results were in good agreement with the test results, and the corresponding error was only 1.3% and 0.7%, respectively. Table 4 lists the comparison of the deflection of the 1# girder at the mid-span under the symmetrical loading test between the test results and the FE analysis. The deflections of the 1# girder at the mid-span from the FE analysis were 4.71 mm and 3.62 mm before and after strengthening, respectively, and they agreed well with the corresponding deflection of the test results.

Figure 16 shows the comparison of lateral load distribution factors between the test and FE analysis under the symmetrical and static partial loading cases before and after ESP strengthening. Under the symmetrical static loading case, the load distribution factors from the FE analysis before strengthening were 0.37, 0.26, and 0.37 for the 1#, 2#, and 3# girders, respectively. After strengthening, the load distribution factors from the FE analysis were 0.345, 0.31, and 0.345 in order. Under the static partial loading condition, the load distribution factors from the FE analysis were 0.1, 0.32, and 0.58 for the 1#, 2#, and 3# girders before strengthening, respectively. After strengthening, the corresponding load distribution factors from the FE analysis were 0.32, 0.33, and 0.35, respectively. This indicates that the FE analysis results were consistent with the field tests’ results.

Table 5 displays the stress contours of the 1# girder before and after ESP strengthening. The maximum longitudinal stress was located in the underside of box girder at the mid-span. The girder strengthened based on the ESP method had a relatively low stress distribution at the same load condition, which indicated that ESP method could be used to improve the capacity of the box girder bridge.

## 5. Conclusions

In this paper, a 26-year-old multi-beam box girder bridge was reinforced by adopting the ESP method, and the strengthening results were investigated by adopting field tests before and after strengthening. The conclusions are drawn as follows.


(1)In comparison to the box girder bridge before strengthening, the field test results showed that the strengthening effect of the deflections and relative displacements under the symmetrical/partial loading case was improved greatly. The maximum concrete strain values under the symmetrical/partial load case were decreased by 23.6%/63% after strengthening, respectively. However, the minimum concrete strain values under the symmetrical/partial load case were increased by 26.7%/61%, respectively. This indicates that the box girder bridge behaves more monolithically after strengthening.(2)The results showed that the first three order frequencies before strengthening were 8.875 Hz, 11.274 Hz, and 18.167 Hz, respectively, and the corresponding frequencies were 17.578 Hz, 25.282 Hz, and 30.416 Hz after strengthening for the background bridge, respectively. The fundamental frequency was increased by 46%, which shows the overall stiffness of the box girder bridge was enhanced.(3)The coefficient of variation of the deflection and strain distribution factors was improved by 53% and 50% under the symmetrical load case for the actual bridge, respectively. The coefficient of variation of the deflection and strain distribution factors was improved by 89% and 84% under the partial load condition, respectively. This presents that the load lateral transferring performance between the box girders is enhanced under two cases.(4)It was manifested that the FE analysis results had a good agreement with the test results for the fundamental frequency, load lateral distribution factor, and deflection. The FE analysis results further verified that the ESP method could be used to improve the mechanical performance of multi-beam box girder bridges.


## Figures and Tables

**Figure 1 materials-15-04353-f001:**
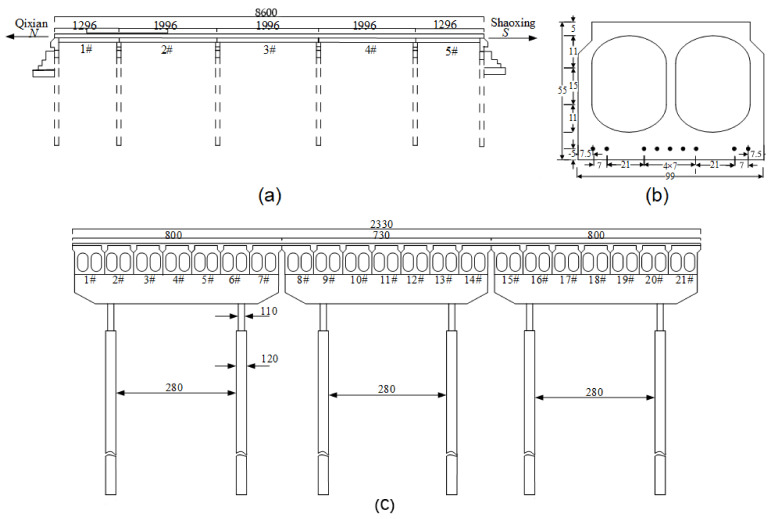
Background bridge (cm). (**a**) Longitudinal section. (**b**) Girder section. (**c**) Transverse section.

**Figure 2 materials-15-04353-f002:**
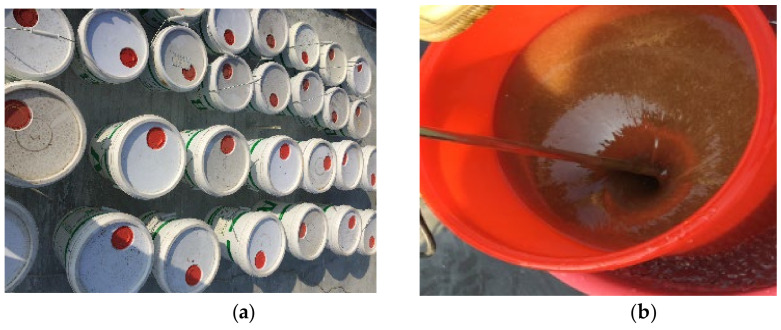
Carbon-A/-B glue and mixture. (**a**) Glue. (**b**) Mixture.

**Figure 3 materials-15-04353-f003:**
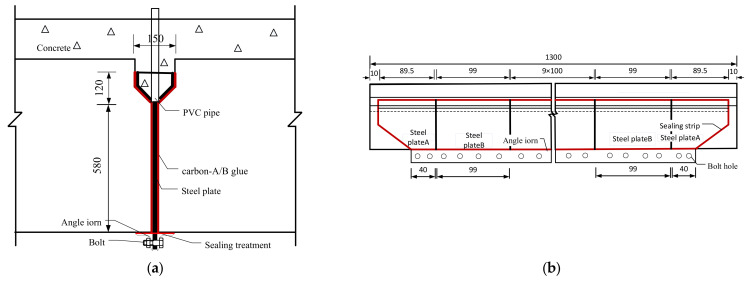
Details of inserting the steel plate. (**a**) Transverse section (mm). (**b**) Longitudinal section (cm).

**Figure 4 materials-15-04353-f004:**
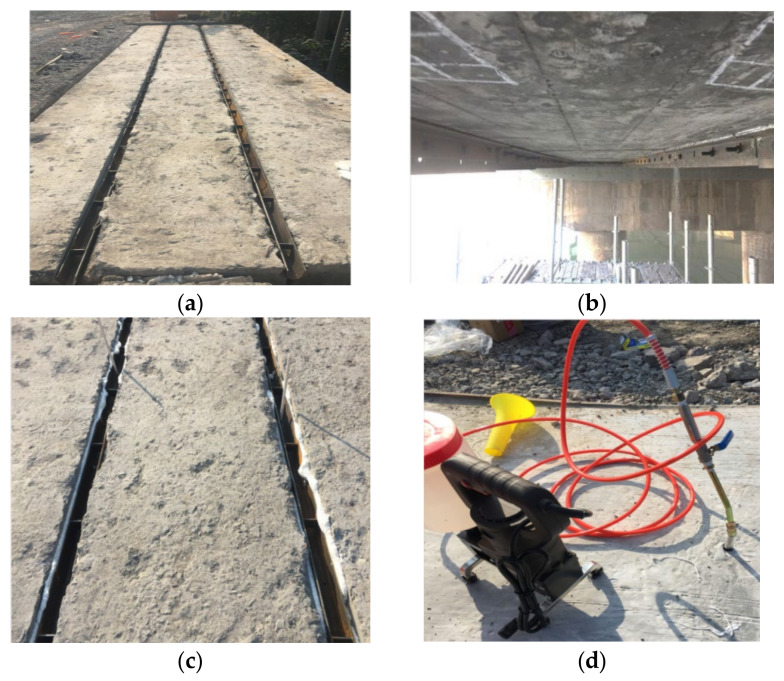
Process of the ESP strengthening method. (**a**) Cleanup and installation. (**b**) Sealing treatment. (**c**) Insertion of PVC pipe. (**d**) Cast-in-place and grout.

**Figure 5 materials-15-04353-f005:**
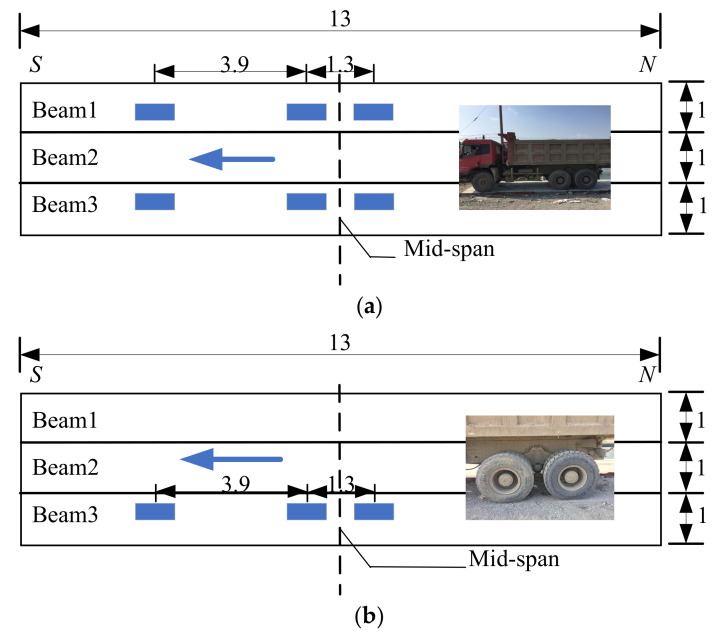
Truck load configuration (m). (**a**) Symmetrical loading. (**b**) Partial loading.

**Figure 6 materials-15-04353-f006:**
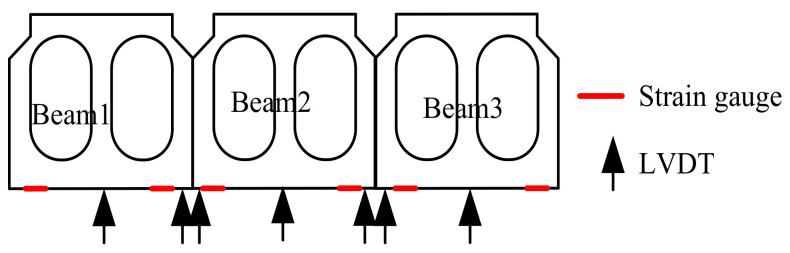
Layout of the strain gauge and LVDTs.

**Figure 7 materials-15-04353-f007:**
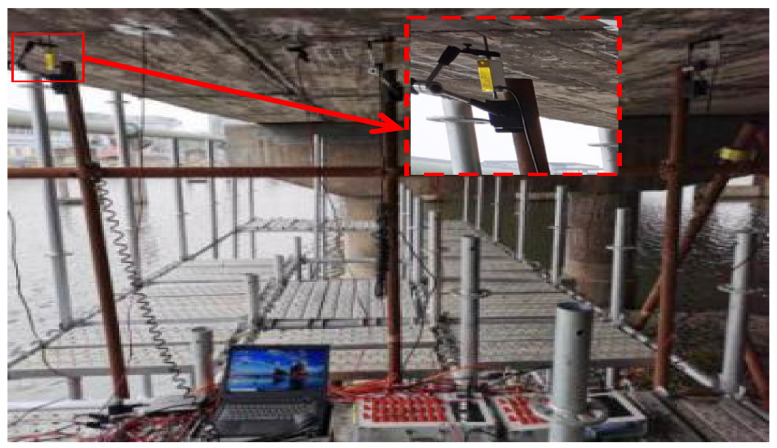
Static test.

**Figure 8 materials-15-04353-f008:**
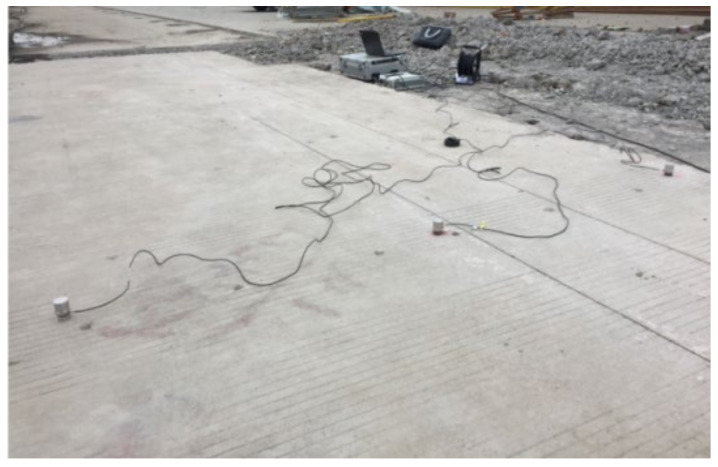
Dynamic test.

**Figure 9 materials-15-04353-f009:**
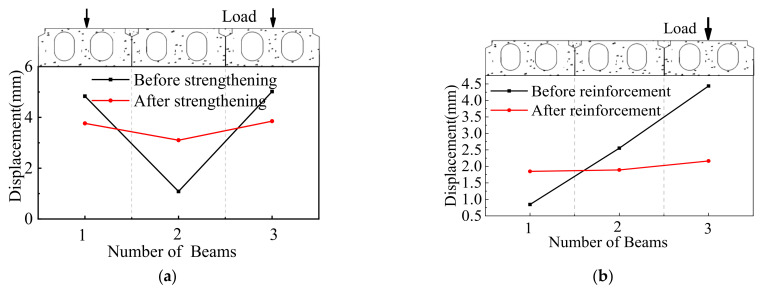
Displacement at the mid-span. (**a**) Symmetrical loading. (**b**) Partial loading.

**Figure 10 materials-15-04353-f010:**
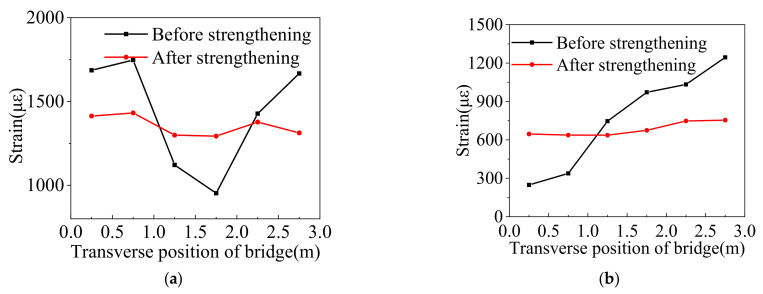
Concrete strains in the underside of the box girders at the mid-span. (**a**) Symmetrical loading. (**b**) Partial loading.

**Figure 11 materials-15-04353-f011:**
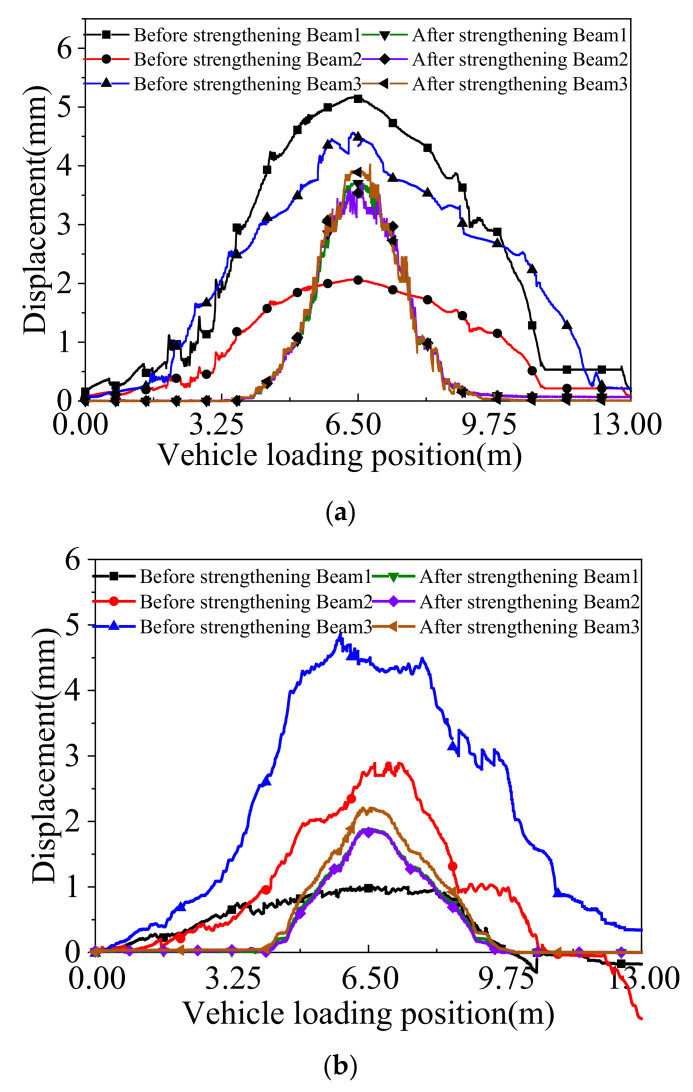
Deflections of the box girder at the mid-span under the continuous loading case. (**a**) Symmetrical loading. (**b**) Partial loading.

**Figure 12 materials-15-04353-f012:**
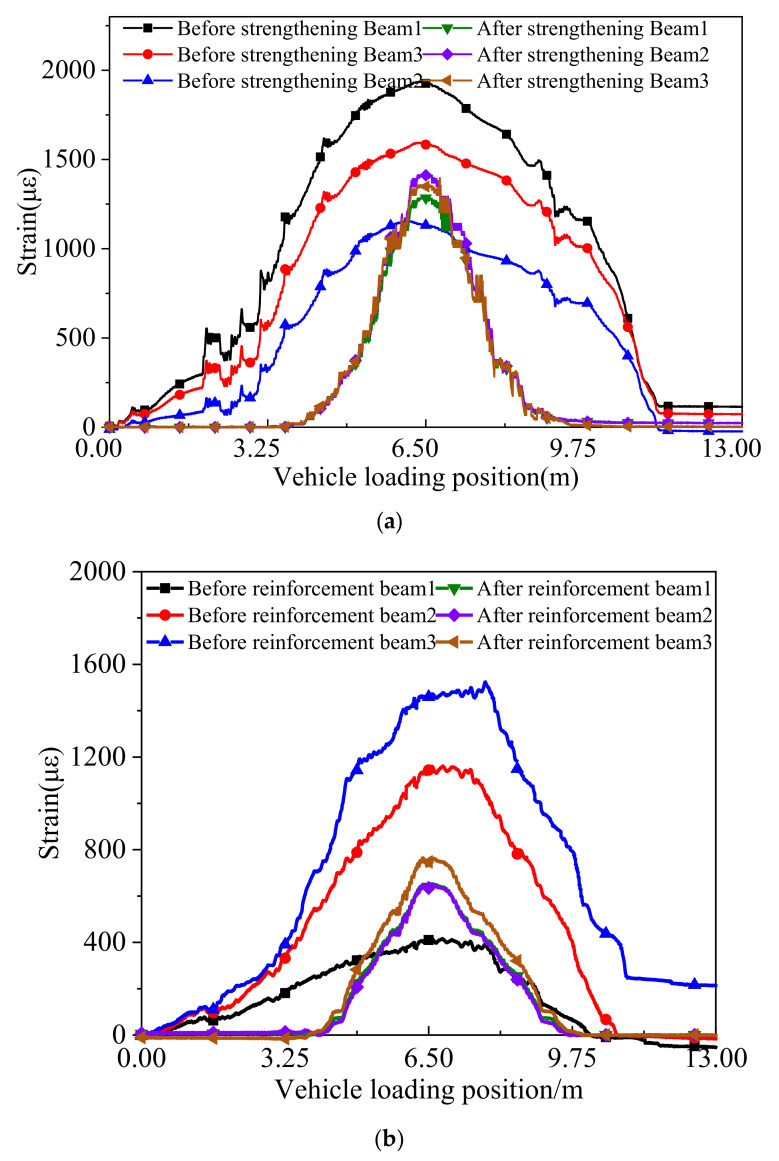
Concrete strain curves of the box girder at the mid-span under continuous loading. (**a**) Symmetrical loading. (**b**) Partial loading.

**Figure 13 materials-15-04353-f013:**
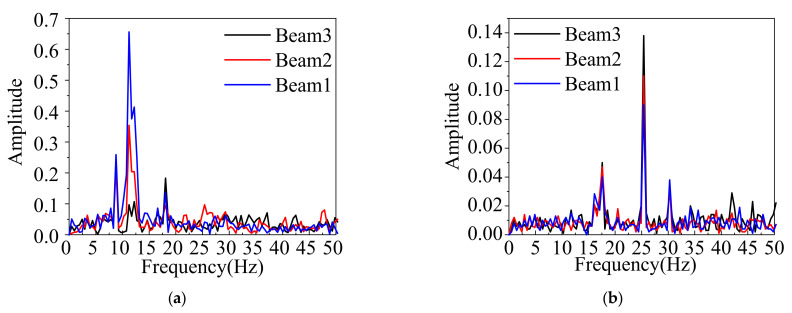
FFT Spectrum. (**a**) Before ESP strengthening. (**b**) After ESP strengthening.

**Figure 14 materials-15-04353-f014:**
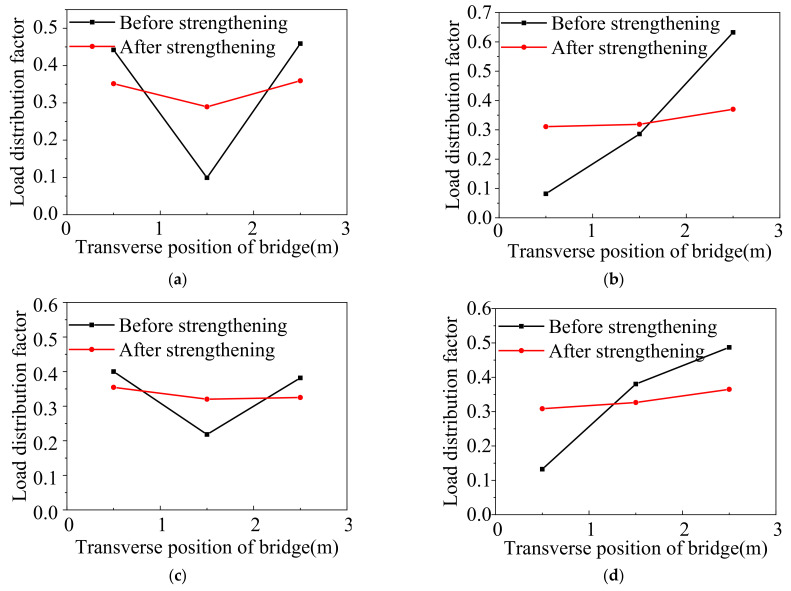
Lateral load distribution factor from field tests. (**a**) Deflection distribution factors under symmetrical load. (**b**) Deflection distribution factors under partial load. (**c**) Strain distribution factors under symmetrical loading. (**d**) Strain distribution factors under partial loading.

**Figure 15 materials-15-04353-f015:**
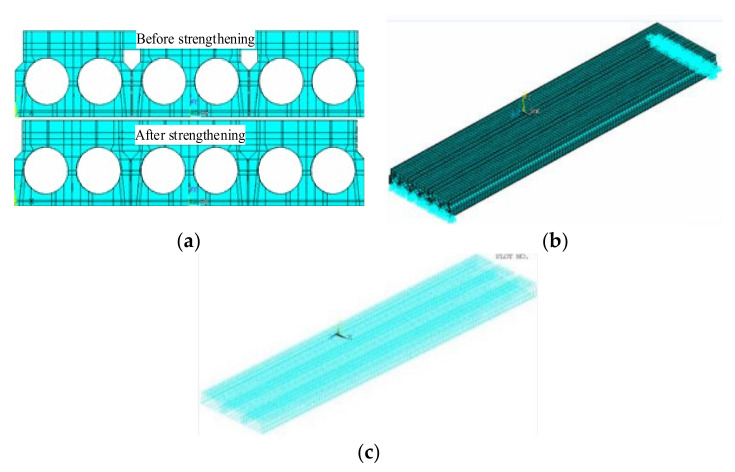
Finite element (FE) model. (**a**) Analysis model. (**b**) Boundary conditions. (**c**) Reinforcement.

**Figure 16 materials-15-04353-f016:**
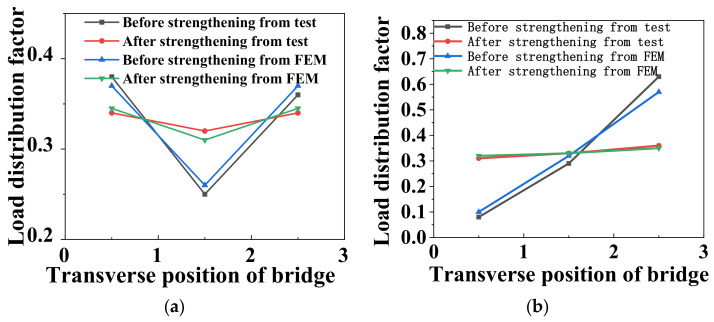
Comparison of load distribution factors between the test and FE analysis. (**a**) Symmetrical loading. (**b**) Partial loading.

**Table 1 materials-15-04353-t001:** Material properties of carbon-A/-B glue and the steel plate.

Shear Strength (Mpa)		Yield Strength (Mpa)	Tensile Strength (Mpa)	Young’s Modulus (Gpa)
Steel plate	265	435	540	210
Carbon-A/-B glue	≥15	≥65	≥30	≥2.5

**Table 2 materials-15-04353-t002:** Load Vehicle Parameters.

Axle	Axle Load (kN)	Left Tire (kN)	Right Tire (kN)	Distance between Axle (m)	Axle Width (m)	Truck Total Load (kN)
1	80	39	41	3.9	2.036	300
2	111	55	56	1.3	1.860	
3	109	53	56	/	1.860	

**Table 3 materials-15-04353-t003:** Fundamental frequency results.

Model Number	Fundamental Frequency (Hz)				Model Shape
	Test		FEA		
	Before	After	Before	After	
1	8.88	17.58	8.76	17.45	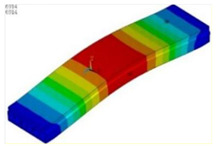

**Table 4 materials-15-04353-t004:** Comparison of the deflection of the 1# girder at the mid-span under the symmetrical load.

Deflection	Before Strengthening	After Strengthening	Before/After (%)
Test (mm)	4.83	3.77	128.1
FE analysis (mm)	4.71	3.62	130.1

**Table 5 materials-15-04353-t005:** Stress contours of the 1# girder under the symmetrical loading condition.

Load Step	Before Strengthening	After Strengthening
100 kN	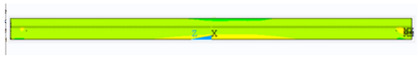	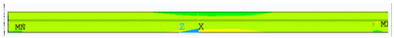
200 kN	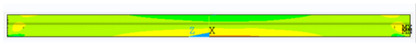	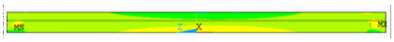
300 kN	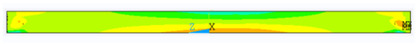	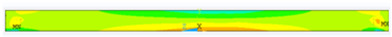

## Data Availability

Some or all data, models, or code that support the findings of this study are available from the corresponding author upon reasonable request.

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
