# Peer review of "Reinforcement Analysis of an Old Multi-Beam Box Girder Based on a New Embedded Steel Plate (ESP) Strengthening Method"

_materials, 2022, doi:10.3390/ma15124353_

Round 1

Reviewer 1 Report

Chennai,

1st June, 2022.

Referee report:

Manuscript ID: materials-1758512

Title: Reinforcement analysis of an old multi-beam box girder based on a new ESP strengthening method

Authors: He Yu Liang, Wang Kai, Cao Zhong Yong, Zheng Pei Juan* and Xiang Yi Qiang

Summary: The work Zheng’s group deals with developing an embedded steel plate (ESP) method comprising of introducing carbon-A/B glue to reinforce longitudinal joins of old multi-beam box girder bridges.    The results have bearing with the safety and durability of the bridges and helps to alleviate the problem of cracks of bridge decks during the service period.   The feasibility of the proposed method for addressing the problem of cracking of longitudinal joints between the box girders is evaluated in the case of real bridges by reinforcing an old multi-beam box girder bridge. Moreover, the structural parameters, namely, strain, frequency, and deflection were examined before and after strengthening with ESP.  Finite element (FE) model of the back ground bridge was setup using ANSYS 18.0 to analyse the strengthening process.  Though a web of Science search with the keywords, namely, multi-beam box girders, yield 3 results, none of them deal with the findings reported by Zheng et al. Owing to the originality and usefulness of results, the work of Zheng’s group is recommended for publication in the journal “Materials” after minor revision.

Major issues: None

Minor issues:  The references needs to be rewritten properly.  In some instances, the references are incomplete; in other instances, the authors names are either incorrectly written or some of the authors names are missing. 

English language need to be improved throughout the paper.

Continuous line numbers to be provided throughout the manuscript that makes the referee to pinpoint the required changes.

Proper spacing between “words” and “numerical values” to be provided.

Professor spacing between “numerical values” and “units” to be provided.

Typographical errors to be corrected.

In addition, the following specific changes can be made for improving the value of the manuscript.

Page number

Revision

1

Title: Replace “ESP” with “embedded steel plate (ESP)”

1

Unbold “Multi-beam”

1

Replace “in the world” with “throughout the world”

1

Replace “before” with “for”

1

Replace “thus affects” with “thus finally affects”

1

Delete “finally” after “bridges”

1

Delete “So in this paper,”

1

What does the “frequency” refers to?

1

Replace “strains, frequencies and deflections” with “strain, frequency and deflection”

1

Replace “by using” with “using” throughout the manuscript

1

Replace “performance” with “process”

1

Replace “ANASYS” with “ANSYS”

1

Insert space between “integrity” and “[1]”

1

Replace “cost-effective” with “cost-effectiveness”

1

Replace “aes-thetic” with “aesthetic”

1

Insert “space” between “appeal” and “[2, 3]”

2

Replace “is lost attraction” with “has lost attraction”

2

Insert “space” between “temperature” and “[4]”

2

Replace “reflect” with “spread”

2

Add “and lead to” before “salt and water leakage”

2

Insert space between “key,” and “and”

2

Replace “Load lateral” with “Lateral load”

2

Insert space between “concerns” and “[5]”

2

Reference number to be provided while citing the work of the authors, namely, Attanayake and Aktan 2015”

2

Replace “postten-sioning” with “post-tensioning”

2

Replace “The results shows that the” with “The results showed that these”

2

Insert space between “England” and “[6]”

2

Insert space between “[7]” and “indicated”

2

After “Attanayake and Aktan 2015” add “[2]”

2

Insert space between “et al.,” and “[7]”

2

Replace “subjects” with “subjected”

16

Reference [7] : replace “T. M. K. Y. T.” with “A”

17

Reference [12]: Replace “H. H. Shad, M. Sargand, Kenneth K Walsh et al.,” with “S. M. Sargand, K. K. Walsh, H. H. Hussein, F. T. A. Rikabi, E. P. Steinberg”

2

Reference [8]: Authors names and reference number do not match

2

Reference [11]: Replace “Hussein et al.,” with “Hussein and Steinberg”

17

Reference [14]: Replace “Z. P. Fu, C. C., M. S. Ahmed.,” with “C. C. Fu, Z. F. Pan, M. S. Ahmed,”

17

Reference [15]: Replace “J. H. E.-E. Huckelbridge, A. A., F. Moses” with “A. A. Huckelbridge Jr., H. El-Esnawi, F. Moses”

16

Reference [1]: Reference [1] could not be retrieved with the information provided.  Complete reference with DOI to be provided

16

Reference [2]: Replace “abc” with “ABC”

16

Reference [2]: Insert “space” between “by” and “side”

16

Reference [4]: Replace “J. E. A. Grace, N. F., M. R. Bebawy,” with “N. F. Grace, E. A. Jensen, M. R. Bebawy”

16

Reference [4]: After “48-63” add “2012”

16

Reference [5]: With the information provided, the reference could not be traced out.  Provide complete reference with doi.

16

Reference [6]: Replace “J. W. N. H. J. K.J. H. K. S. B. K. Kim, J. J., K. J. Byun,” with “J. H. J. Kim, J. W. Nam, H. J. Kim, J. H. Kim, S. B. Kim, K. J. Byun”

17

Reference [8]: Replace “L.  Zhengyu, P. B. M.” with “Z. G. Liu, B. M. Phares”

17

Reference [8]:  Add “24 (12), 2019,” after “Bridge Eng.”

17

Reference [11]: Replace “W. K. S. S. Hussein, H. E. Steinberg” with “H. H. Hussein, K. K. Walsh, S. M. Sargand, E. P. Steinberg, K. K. Walsh, S. M. Sargand, E. P. Steinberg”

17

Reference [11]: Add “28(5)” before “(2016)”

5

The font size and font type of the text in Table 1 are different from the rest of the manuscript

5

In section 2.3, the values “540

                                           ≥ 30”  to be deleted

8

The font size and font type of the text in Table 2 are different from the rest of the manuscript

13

Caption of Figure 15: Replace “FE” with “Finite element (FE)”

13

The font size and font type of the text in Table 3 are different from the rest of the manuscript

14

The font size and font type of the text in Table 4 are different from the rest of the manuscript

14

Figure 16: The labelling of X and Y axis and the text in figure were unclear and the font size is too small to be read

15

The font size and font type of the text in Table 5 are different from the rest of the manuscript

2

Insert space between “Phares” and “[8]”

2

Replace “shrink-age” with “shrinkage”

2

Replace “appear cracks” with “have cracks”

2

Replace “rela-tive” with “relative”

2

Insert space between “Sang” and “[9]”

2

Replace “were not” with “have not”

2

Replace “but did” with “but did occur”

2

Replace “These showed” with “This showed”

2

Insert space between “Graybeal” and “[10]”

2

Insert space between “Hussein et al.,” and “[11]”

2

Replace “Hus-sein” with “Hussein”

2

Delete “the” before “high adhesion”

2

Replace “increases with” with “increased with”

2

Insert space between “Shad et al.,” and “[12]”

2

Replace “in all of the” with “of all of the”

2

Replace “American” with “America”

2

Insert space between “America” and “[5]”

2

Replace “at early years” with “during early years”

2

Add “and” before “about 80 %”

2

Insert space between “12” and “m”

“12m” to be written as “12 m”

2

“18m” to be written as “18 m”

2

Add “are” before “replaced”

2

Insert space between “year” and “[13]”

2

Replace “The replaced cause” with “The cause for the replacement”

2

Rewrite “con-cerns” as “concerns”

2

Delete “for” before “most”

2

Delete “causing” with “inconvenient”

2

Replace “authors” with “author’s”

2

Replace “researches” with “studies”

2

Insert space between “Aktan” and “[2]”

2

Rewrite “posttensioning” as “post-tensioning”

Apply this change throughout the manuscript

2

Replace “make” with “made”

2

Replace “be no good” with “ineffective”

2

Insert space between “Fu et al.,” and “[14]”

3

Replace “trans-fer” with “transfer”

3

Replace “mit-igate” with “mitigate”

3

Insert space between “cracking” and “[9, 10]”

3

Delete “The” before “Carbon-A/B”

3

Add “gives” before “good mobility”

3

Replace “re-duce” with “reduce”

3

Replace “gird-ers” with “girders”

3

Replace “monolithically” with “monolithic”

3

Replace “three-dimension FE” with “three dimensional finite element (FE)”

3

Delete “also” before “developed”

Reviewer 2 Report

Review report on the topic ‘Reinforcement analysis of an old multi-beam box girder based on a new ESP strengthening method’. Comments are listed below:

1.        Strengthen the abstract section. Add the key conclusion of the works in the last two lines of the abstract section.

2.        Discuss the motive behind the work. The clear application of the work should be discussed in the introduction section. Such type of dissimilar joint application should be discussed first in the introduction section. From the introduction section application of the work is not clear.

3.        There are numerous spelling and grammatical errors. Please revise the manuscript thoroughly. Sentences are also not complete.

4.        The novelty of the work should also be discussed in a separate paragraph.

5.        Experimental section need more discussion.

6.        The discussion section is very poor. The results are presented like technical report. Add more discussion with proper references.

7.        Shorten the length of the conclusion section. Add key points only.

Reviewer 3 Report

The Author has done the analysis of multi beam box girder based on ESP strengthening Method. It is really a innovative technique. The title of the manuscript is appropriate to the content. The introduction part explains the background of the research and referred sufficient literature. The experimental part is written in a good manner with sufficient supporting figures. All the Results are presented and discussed in section 3. Section 4 explains the FEA and the results are compared with the tests. The conclusion is well supported by results obtained from the study.

Overall the Manuscript is good and can be accepted with minor revision after carrying out the following correction.

·        Title should not have acronyms. So, give full form to ESP in the title.

·        Increase the number of keywords.

·        Provide the clear image of figure 1. Especially Fig. 1 (b).

·        The dimensions Fig 3 is not clear. If possible, present a clear image.

·        The legend of Fig 11,12,13,14,16 is not visible. Revise it.

·        Section 4. Finite element analysis has smaller font size. Increase it as per journal guidelines Table 3 is not as per journal format.

·        Conclusion section has smaller font size. Increase it as per journal guidelines

·        Can increase the number of citations..

Reviewer 4 Report

REVIEW

on article

Reinforcement analysis of an old multi-beam box girder based on a new ESP strengthening method

HE Yu Liang, WANG Kai, CAO Zong yong, ZHENG Pei juan and XIANG Yi qiang

SUMMARY

The article submitted for review is devoted to the current topic: "Calculation of the reinforcement of an old multi-beam box-shaped beam based on a new method of reinforcing with an embedded steel plate." The topic is relevant, since multi-beam bridges with box girders have been widely used in the world for many years. However, there is a problem, which is identified in the study as follows: cracking of longitudinal joints between box-section beams always leads to reflective cracks in the bridge deck during operation, thus affecting the safety and durability of the bridges themselves. The article proposes an interesting original method for solving the problem - this is the method of strengthening by introducing carbon glue A / B to strengthen the longitudinal joints of old multi-girder bridges with box girders. The authors used innovative research methods, tested the result and obtained a number of important data.

The results of the analysis showed that the structural rigidity and the efficiency of transverse load transfer between the box beams were improved after reinforcement.

Thus, the article is original, has a specific degree of novelty and practical significance. However, there are shortcomings in the article that need to be corrected. They are listed below.

COMMENTS

1.    The Abstract needs to be finalized, supplementing it with the results obtained. In the presented form, the Abstract contains a few results achieved. 

2. Is it correct that author's names are written in small letters?

3.    The Abstract is presented somewhat vaguely, it is necessary to specify the obtained scientific results by adding their quantitative characteristics.

4.    The "Introduction" section is written too concisely and needs to be improved. To fully assess the scientific novelty and quality of the literature review, this section should be enlarged. In addition, the small number of analyzed sources is striking; the number of sources in the literature review should be increased to 20-25 pieces.

5.    The ending of the "Introduction" section looks somewhat ponderous due to the continuous text. Perhaps some visualization elements should have been added to make the article more easily perceived by readers.

6.    Section 2 in subsection 2.1 presents Figure 1 with fragments a, b, c. These fragments are presented in very low quality, are unreadable, these figures should be presented in proper quality. In addition, it is necessary to give a more detailed explanation of these figures.

7.    It is methodologically incorrect to complete a subsection with a figure, it should be followed by a textual interpretation with a clear division, which is shown in Figure 1a, which is shown in Figure 1b and which is shown in Figure 1c.

8.    The beginning of subsection 2.2 "Material properties" should be supplemented with some preamble. The section begins by suggesting that carbon glue is a kind of composite material, as showed in Figure 2. It should be explained why this material was chosen.

9.    Figure 2 and others have an awful quality; it should be brought into line with the quality of the image and the design of the figures: 1000 pix for the shorten side and 300 DPI resolution.

10.    Subsection 2.2 ends with a table that needs, firstly, a textual interpretation, and secondly, bringing this interpretation after the table itself.

11.  In subsection 2.3 there are typos, you should check the format of the text and figures. In addition, Figure 3a,b is invalid due to the illegibility of the characters in these figures.

12.  Figure 4a,b,c,d needs further clarification and improvement in the quality of the image presentation. Similar comments apply to subsection 2.4 and Figures 5, 6, 7, 8.

13.  The experimental results in Section 3 are presented rather concisely, the graphs in Figures 9 and 10 are not very informative and need more detailed explanation and textual interpretation.

14.  Graphs in Figures 11, 12 and 13 are of very poor quality, are unreadable and unacceptable for publication in their current form, quality drawings should be provided. At the moment it is impossible to distinguish what is shown in the figures.

15.  Formula 1 is in an invalid format and needs to be properly formatted. Also, there are no sufficient explanations for this formula. It is unacceptable to leave it in this form.

16.  Figure 14 is also invalid in terms of image quality.

17.  Section 4 presents a large number of graphical elements that are not properly explained. Detailed textual interpretation should be provided and smooth transitions between sections and figures should be provided. In addition, section 4 ends with a table, which is also methodologically incorrect.

18.  The "References" section includes only 17 sources that need to be increased to 20-25 sources, at least.

19.  The next remark on the list of references is the lack of a sufficient number of sources published over the past 5 years. The literature review should be supplemented with fresh sources in order to talk about scientific novelty. The incorrect design of literature sources is striking.

20.  The section "Conclusions" is rather cumbersome. It is necessary to point out the main results achieved in the study and present the formulations of their practical significance and further promising areas of research.

21.  In general, the conclusion of the reviewer on the article. The study is quite interesting and promising, however, in its current form, the article cannot be published in the Journal «Materials». A complete check of the style is required, an addition from the point of view of literature and the author's own interpretations, as well as a complete revised of the design style and an increase in the quality of all images: both tabular and graphic elements, and formulas. After a serious correction, the article must be re-reviewed.

Round 2

Reviewer 2 Report

Accepted. 

Reviewer 4 Report

All my comments were taken into account and appropriate corrections were made in the article's text. I recommend the article for publishing.